# Prebiotic Properties of Exopolysaccharides from *Lactobacillus helveticus* LZ-R-5 and *L. pentosus* LZ-R-17 Evaluated by In Vitro Simulated Digestion and Fermentation

**DOI:** 10.3390/foods11162501

**Published:** 2022-08-18

**Authors:** Mengjia Xu, Zhi Li, Xiaogan Zhao, Wei Li

**Affiliations:** 1State Key Laboratory of Dairy Biotechnology, Shanghai Engineering Research Center of Dairy Biotechnology, Dairy Research Institute, Bright Dairy & Food Co., Ltd., Shanghai 200436, China; 2College of Food Science and Technology, Nanjing Agricultural University, Nanjing 210095, China

**Keywords:** *Lactobacillus helveticus* LZ-R-5, *L**. pentosus* LZ-R-17, exopolysaccharides, in vitro digestion and fermentation, selectivity index, prebiotic

## Abstract

The in vitro digestion and fermentation behaviors of *Lactobacillus helveticus* LZ-R-5- and *L**. pentosus* LZ-R-17-sourced exopolysaccharides (LHEPS and LPEPS) were investigated by stimulated batch-culture fermentation system. The results illustrated that LHEPS was resistant to simulated saliva and gastrointestinal (GSI) digestion, whereas LPEPS generated a few monosaccharides after digestion without significant influence on its main structure. Additionally, LHEPS and LPEPS could be consumed by the human gut microbiota and presented stronger bifidogenic effect comparing to α-glucan and β-glucan, as they promote the proliferation of *Lactobacillus* and *Bifidobacterium* in cultures and exhibited high values of selectivity index (13.88 and 11.78, respectively). Furthermore, LPEPS achieved higher contents of lactic acid and acetic acid (35.74 mM and 45.91 mM, respectively) than LHEPS (35.20 mM and 44.65 mM, respectively) during fermentation for 48 h, thus also resulting in a larger amount of total SCFAs (110.86 mM). These results have clearly indicated the potential prebiotic property of EPS fractions from *L**. helveticus* LZ-R-5 and *L**. pentosus* LZ-R-17, which could be further developed as new functional food prebiotics to beneficially improve human gut health.

## 1. Introduction

Currently, researchers are focusing on the exploration of novel functional foods that can regulate the composition of human intestinal microbiota and further help to maintain the homeostasis of physiological processes [1,2]. Bifidobateria and lactobacilli, the main microorganisms beneficially influencing the host’s health, are motivated by prebiotics [3]. As a result, prebiotics represent one of the substrates with high availability for modulating a diverse and healthy microbial ecosystem. Additionally, the fermentation reactions of prebiotics would to induce desirable productions including short-chain fatty acids (SCFAs), which can also improve colonic and systemic health [4]. In light of the latest scientific and clinical developments, a prebiotic is redefined as “a substrate that is consumed by host microorganisms selectively and confers health benefits” [5]. They are generally non-digestible naturally-sourced carbohydrates such as fructans and galactans, including fructooligosaccharides (FOS), inulin, and galactooligosaccharides (GOS) [5,6]. However, the prebiotic properties of lactic acid bacteria (LAB)-sourced exopolysaccharides (EPS) are rarely reported. Fermentative, Gram-positive, and non-spore-forming LAB play significant roles in the traditional dairy industry, agriculture, pharmaceutics, and medicine [7]. Most LAB, including *Lactobacillus rhamnosus*, *L. delbrueckii* ssp. *Bulgaricus*, *L. plantarum*, *L. helveticus*, *Streptococcus thermophilus*, and *Lactococcusc lactis* are capable of synthesizing a wide range of EPS [8,9,10]. Due to their selective host cell protection and biological beneficial properties of antimicrobial, anti-inflammatory, anti-tumor, and immunomodulatory activities, EPS from LAB have attracted increased attention in recent years [11,12,13]. Nevertheless, several studies indicated that the chemical composition, surface morphology, and biological activity of polysaccharides could be changed remarkably after simulated saliva and gastrointestinal (GSI) digestion [14,15]. Since prebiotics are non-digestible carbohydrates and would selectively enhance the activity of some groups of beneficial bacteria, the fermentable properties of indigestible polysaccharides by gut microorganisms become a subject with great value. Furthermore, the level of SCFAs that are produced from the fermentation of indigestible polysaccharides is closely related to the prebiotic effect on the host [16]. Therefore, it is crucial to determine the potential mechanism of digestion and fermentation as well as structure-bioactivity relationships of EPS.

Nowadays, in vitro pH-controlled, sterile stirred batch-culture digestion and fermentation systems are extensively acknowledged to simulate environmental conditions in human buccal and intestine [14,17]. The enzymes, bile salts, and varied pH during the digestion phase would affect the structural and biological properties of polysaccharides [18]. Thus, it is a satisfied model to provide a thorough understanding of the digestion and fermentation behaviors of polysaccharides by simulated saliva and GSI conditions and complex intestine bacterial, which would further offer preliminary proof for the prebiotic effects of the studied EPS on promoting host health. Moreover, inoculating human gut microbiota samples as opposed to pure-, mixed-, or co-cultures is a more convincing approach to ensure the diversity of colonic microflora [19].

Recently, two strains of LAB (*L. helveticus* LZ-R-5 and *L. pentosus* LZ-R-17) with a relative higher yield of EPS were isolated from Tibetan kefir grains in our laboratory. There were two EPS fractions named LHEPS and LPEPS that were obtained and separated by DEAE-52 cellulose anion-exchange column. The structural characteristics and immunomodulatory capacity of these two EPS fractions were investigated and revealed [20,21]. The results showed that both of them were heteropolysaccharides and LPEPS presented a larger average molecular weight (Mw) than LHEPS (1.20 × 10^6^ Da and 5.41 × 10^5^ Da, respectively). In addition, these two EPS fractions were composed of galactose and glucose with a molar ratio of 1.00:3.15 and 1.00:3.65. The LPEPS contained repeating units of →2)-α-D-Gal*p*-(1→4)-β-D-Glc*p*-(1→4)-β-D-Glc*p*-(1→4)-β-D-Glc*p*-(1→ and the LHEPS was composed of →6)-β-D-Gal*p*-(1→3)-β-D-Glc*p*-(1→3)-β-D-Glc*p*-(1→3)-β-D-Glc*p*-(1→3)-β-D-Glc*p*-(1→ [20,21]. In this study, the dynamic changes of physicochemical properties of purified LHEPS and LPEPS fractions during the in vitro stimulated digestion, as well as their fermentability and prebiotic function in human colonic microbiota were investigated. Besides, the variations of Mw, structural integrity, release of dissociated monosaccharides or oligosaccharides, and the production of SCFAs were quantitatively analyzed to assess the availability of EPS by the gut bacteria. Furthermore, the influence of EPS intake on the composition of human gut microflora were conducted by fluorescence in situ hybridization (FISH). This study provides the latest valuable information on the mechanisms of absorption and utilization of EPS from LAB in the buccal and gastrointestinal tract of human body.

## 2. Materials and Methods

### 2.1. Materials and Chemicals

Amyloglucosidase, pancreatic α-amylase, salivary α-amylase, bile salt, pancreatin, resazurin, hemin, vitamin K_1_, L-cysteine, α-glucan, β-glucan and short-chain fatty acids standards, 3-methyl-1-phenyl-2-pyrazolin-5-one (PMP), glucose (Glc), galactose (Gal), rhamnose (Rha), mannose (Man), arabinose (Ara), Coomassie Brilliant Blue G-250 dye, bovine serum albumin (BSA), and 2-(4-Amidinophenyl)-6-indolecarbamidine dihydrochloride (DIPA) staining solution were obtained from Sigma–Aldrich Chemical Co. (St. Louis, MO, USA). Pepsin (>400 U/mg) and trypsin (250 U/mg) were obtained from Fluka (Buchs, Switzerland). Inulin (source: chicory, content ≥ 95%) from Yuanye Bio-Technology Co., Ltd. (Shanghai, China) has a molecular weight of around 5000 Da. T-series dextrans were purchased from Pharmacia Co., Ltd. (Uppsala, Sweden). All other reagents that were utilized in this study were of analytical grade.

### 2.2. Isolation and Purification of LHEPS and LPEPS

The LHEPS and LPEPS were produced by *L. helveticus* LZ-R-5 and *L. pentosus* LZ-R-17 strains that were isolated from Tibetan kefir (gathered from common families in the Nyingchi, Tibet of China) in our laboratory [20,21]. Cultivations of *L. helveticus* LZ-R-5 and *L. pentosus* LZ-R-17 were carried out as batch cultures for the production of EPS. Precisely, the optimal inoculum concentration was 4% (*v*/*v*) in pasteurized milk (sourced from cow, Bright Dairy & Food Co., Shanghai, China) and the culture mediums were kept at 37 °C for 24 h, respectively. The supernatant was then collected by centrifugation for 25 min at 12,000 rpm and 4 °C with the addition of 4% (*w*/*v*) trichloroacetic acid (TCA) and kept for another 6 h to remove denatured proteins. After centrifugation, the clear supernatant was collected and concentrated using a rotary evaporator (Rotavapor^®^ R-250 Pro, BUCHI, Flawil, Switzerland). Then, three volumes of ice-cold 98% ethanol were added to the precipitate the EPS for 12 h at 4 °C followed by centrifugation at 12,000 rpm for 15 min (4 °C). Afterwards, the precipitate was suspended in distilled water and dialyzed for 3 days at 4 °C. The crude LHEPS and LPEPS (10 mL, 10 mg/mL) were collected by further concentration and lyophilization, as well as purified by a DEAE-52 anion exchange column (2.6 × 30 cm) that was purchased from Whatman Co., Ltd. (Maidstone, Kebt, UK). The elution was conducted with a step gradient of 0.0, 0.1, and 0.3 M NaCl/H_2_O at 1 mL/min flow rate.

For the basic chemical component analysis, the total sugar content and protein content were measured by the phenol-sulfuric acid method and the Bradford method as described previously [22,23]. For the measurement of the total sugar content, 2 mL of sugar solution was mixed with 0.05 mL of 80% phenol (*w*/*w*) and then 5 mL of concentrated sulfuric acid was added rapidly. After standing for 10 min, the mixture was shaken and placed for 5 min in a water bath at 80 °C. The absorbance of the mixture was measured at 490 nm (UV-vis spectrophotometer, Cary 300, Agilent, Palo Alto, CA, USA) and the carbohydrate concentration was determined according to a standard curve. For the analysis of the total protein content, 5 μL of the sample and 250 μL of the Bradford reagent were added to the wells and the 96-well plate was incubated at 37 °C for 5 min, after which the absorbance of the solution was measured using a microplate spectrophotometer (SpectraMax 190, Molecular Devices, Sunnyvale, CA, USA) at 595 nm. The Bradford reagent was prepared by mixing accurate 100 mg Coomassie Brilliant Blue G-250 dye with 50 mL 95% ethanol. Then, 100 mL phosphoric acid was added and the mixture was diluted to 1 L. Stock BSA solutions were used to prepare calibrators. Additionally, the total uronic acid content and sulfate group content were determined according to previous methods [24,25,26]. Briefly, to determinate the total uronic acid content, 1.2 mL of sulfuric acid/tetraborate solution was added to 0.2 mL of sample solution, and the mixture was shaken vigorously and heated in a water bath at 100 °C for 5 min. After cooling on ice, 20 μL of the *m*-hydroxydiphenyl reagent was added. The mixture was shaken gently, and the absorbance was read at 520 nm within 5 min. Furthermore, the sulfate group content was determined based on a pyrolytic method. Briefly, the samples or sulfate standards were mixed with 5 μL of 0.02 M NaOH and the dried residues were pyrolyzed in ignition tubes by heating evenly for approximately 6 s in a Fisher burner. Then, the residues were dissolved in 0.25 mL deionized water with the addition of barium buffer (0.6 mL) and rhodizonate reagent (0.3 mL). After mixing and incubating for 10 min at room temperature, the absorbances were measured at 520 nm.

The molecular weight and monosaccharide composition analysis of purified LHEPS and LPEPS was determined by high-performance liquid chromatography (HPLC, Agilent 1100 series, Palo Alto, CA, USA) as described in Section 2.4. For determination of the functional groups in LHEPS and LHEPS, Fourier-transform infrared spectroscopy (FT-IR, Thermo Fisher Nicolet iS50, Waltham, MA, USA) from 400 to 4000 cm^−1^ was recorded at a resolution of 4 cm^−1^ with 16 scans. The structure of LHEPS and LHEPS was performed with D_2_O as the solvent by ^1^H NMR, ^13^C NMR and 2D NMR spectra (Bruker AVANCE AV-500, Bruker Group, Fällanden, Switzerland). The spectrometer was operating at 500 MHz with a temperature of 323 K. For the ^1^H NMR and ^13^C NMR spectra, the delay (*D*_l_) and acquisition (AQ) times were 4.00 and 2.92 s, 1.08, and 2.00 s, respectively. The 2D ^1^H-^1^H correlated spectroscopy (COSY), total correlation spectroscopy (TOCSY), ^1^H-^13^C heteronuclear single quantum coherence (HSQC), ^1^H-^13^C heteronuclear multiple quantum coherence (HMBC), and nuclear Overhauser effect spectroscopy (NOESY) measurements were used to determine the sequence of sugar residues.

### 2.3. In Vitro Simulated Saliva and GSI Digestion of LHEPS and LPEPS

The in vitro digestion of LHEPS and LPEPS was performed by our previous methodologies with minor modifications [27]. The procedure was initiated with a simulated human salivary phase, followed by a gastric phase and an intestinal phase. A pH-meter model 744 (Metrohm AG, Herisau, Switzerland) was applied for monitoring the pH of the digestion solution. For the buccal phase, 10 mg of LHEPS or LPEPS was added to the 4 mL mixture of salivary α-amylase and amyloglucosidase, and 10 mL buffer solution. The digestion solution was carried out in a shaking water bath (MaxQ™ 7000, Thermo Fisher, Waltham, MA, USA) for 10 min (37 °C, 55 rpm). For the gastric phase, the pH of digestion solution was adjusted to 2.0 using 1 M HCl, then pepsin was added to a final concentration of 3.0 mg/mL and incubated at 37 °C for 2 h in darkness while stirring (55 rpm). For the intestinal phase, 4 mL mixture of pancreatin, trypsin, pancreatic α-amylase, and bile salt solution was added and 1 M NaOH was applied to regulate the pH of the digestion solution. The digestion phase was continued at 37 °C, 150 rpm for 8 h. Finally, the mediums were inactivated immediately at 100 °C for 5 min and filtered before analysis. Each in vitro simulated saliva and GSI digestion was conducted three times or more to ensure repeatability.

### 2.4. Mw Determination and Monosaccharide Composition Analysis

After stimulated digestion, the molecular weight, digestion rate, and monosaccharide composition of LHEPS and LPEPS samples were measured as described previously [20]. For the analysis of the molecular weight, the digestion products (20 μL) and unfermented polysaccharide samples were injected into an HPLC system that was equipped with a TSK GEL G4000 PWXL column (300 × 7.8 mm, Tosoh Corp., Tokyo, Japan) at a flow rate (DI water) of 0.8 mL/min (30 °C), and the linear curve was calibrated with standard T-series dextrans. The monosaccharide composition of EPS was analyzed by an HPLC system with an Eclipse Plus C_18_ column (Agilent, Palo Alto, CA, USA). In brief, 5 mg of EPS sample was hydrolyzed with 2 M trifluoroacetic acid (TFA, 2 mL) at 120 °C for 2 h. After co-distilling with methanol repeatedly, the excess TFA was converted to its PMP derivative. Then, the hydrolysate was dissolved by distilled water and mixed with PMP (0.5 M, 0.4 mL) and NaOH (0.3 M, 0.2 mL) solution for reacting 30 min at 70 °C. The solution was neutralized with 0.3 M HCl followed by bleaching with chloroform for three times to remove excess PMP. The upper aqueous phase product was filtered through a 0.22 μm membrane and analyzed by HPLC with a flow rate of 0.8 mL/min (mobile phases: 0.1 M ammonium acetate (pH = 5.0), acetonitrile, and tetrahydrofuran in a ratio of 81:17:2 (*v*/*v*/*v*)). The monosaccharide components of LHEPS and LPEPS were quantitatively analyzed by comparing the peak time and peak area of each standard monosaccharide derivative (mannose, arabinose, rhamnose, glucose, and galactose) and relative EPS hydrolyzed derivatives.

### 2.5. Origin of Human Fecal Samples and Fecal Suspension Preparation

A total of three healthy human volunteers of two females and one male living in Nanjing, China (ages from 25 to 30) provided the fresh fecal samples. Before participating in this trial, these volunteers had no prior history of gastrointestinal disorders, and had not received any antibiotic and pro- or pre-biotic therapy as well as had not consumed any dairy products containing probiotics for at least half a year. The fecal suspension was made by mixing the freshly obtained fecal samples (1:5 (*w*/*v*)) with anaerobic phosphate buffer solution (1.0 M, pH 7.4) within 24 h after defecation.

### 2.6. In Vitro Fermentation

The purified EPS components were fermented in vitro according to the reported method with minor modifications [28,29]. In brief, the fermentation medium consisted of yeast extract, peptone water, L-cysteine hydrochloride, K_2_HPO_4_, NaHCO_3_, NaCl, CaCl_2_·6H_2_O, MgSO_4_·7H_2_O, bile salt, hemin, Tween 80, vitamin K_1_, and resazurin solution, and the medium pH was adjusted to 7.4 with 1.5 M HCl. A total of 10 mL of each medium was sterilized with the addition of 1% (*w*/*v*) filtered pure LHEPS and LPEPS fractions. Then, following the addition of 10 % (*w*/*w*) freshly prepared fecal suspension as an inoculant, the fermentation medium was incubated for 48 h at 37 °C in an automatic controlled atmosphere chamber (888-simplicity; PLAS LABS, Inc., Lansing, MI, USA). For the records of each bacterial population and the determination of SCFAs, liquid was sampled at 0, 6, 12, 24, and 48 h during the fermentation phase. Each in vitro fermentation was conducted three times to ensure repeatability.

### 2.7. Determination of Bacterial Populations by FISH

One of the most rapid and practical methods for bacteria enumeration is the FISH technique, which employs 16S rRNA-targeted fluorophore-labeled oligonucleotide probes and confocal laser scanning microscopy [30]. Specifically, 300 μL filtered paraformaldehyde was added to 100 μL collected fermentation samples for fixation (10 h, 4 °C). After being centrifuged for 5 min, the bacterial cells were washed twice with 1 mL of sterile filtered PBS and resuspended in 300 μL of PBS/ethanol (1:1, *v*/*v*) and stored at −20 °C for further hybridization.

The 16S rRNA-targeted DNA probes that were labelled with cyanine-3 (Cy3) fluorescent dye (50 ng/μL stock solution) were customized by JIE LI Biology Co., Ltd. (Shanghai, China) for the numeration of the corresponding bacterial groups in mixed cultures (sequences details in Figure 1). While the nucleic acid stain DAPI was applied for the total bacterial counts. Precisely, probe Bif164, Lab158, Str493, Bac303, His150, and EC1531 were specific for *Bifidobacterium* spp., *Lactobacillus*/*Enterococcus* spp., *Streptococcus*/*Lactococcus* ssp., *Bacteroides-Prevotella* group, most species of the *Clostridium histolyticum* group, Clostridium clusters I and II, and *Escherichia coli*, respectively. Bacterial cells were counted on a TCS SP8 laser scanning confocal microscopy (Leica, Wetzlar, Germany). At least ten random fields were counted for each well, and the bacterial numbers were listed as log_10_ cells per milliliter ± standard deviation (SD).

### 2.8. Selectivity Index (SI) Scores

The SI score represents a general quantitative comparative change in the “beneficial” bacteria population (*Bifidobacteria*, *Lactobacillus*/*Enterococcus* spp., and *Streptococcus*/*Lactococcus* spp.) and “undesirable” ones (*Bacteroides-Prevotella* group, *Clostridium histolyticum* group, and *Escherichia coli*), which is also defined to compare the influence of the different oligosaccharides on the selectivity of fermentation [31]. The SI score was calculated according to previously reported equation as follows [32]:(1)$$SI=\frac{\left(Bif_{t}/Bif_{0}\right)+\left(Lab_{t}/Lab_{0}\right)+\left(Str_{t}/Str_{0}\right)-\left(Bac_{t}/Bac_{0}\right)-\left(His_{t}/His_{0}\right)-\left(EC_{t}/EC_{0}\right)}{Totalcount_{t}/Totalcount_{0}}$$
where *Bif_t_*/*Bif*_0_ represents the ratio of *Bifidobacterium* spp. populations at each sampling time point to the initial populations at inoculation, as well as for *Lab*, *Str*, *Bac*, *His*, *EC*, and *Totalcount*. In this equation, “beneficial” bacteria are positive, whereas the rest of the studied species are negative. However, it does not mean that all of the undesirable bacteria groups are necessarily harmful for the balance of the gut ecosystem.

### 2.9. SCFAs Analysis

The contents of SCFAs in the human gut microbiota cultures were measured by an HPLC system as described previously [27]. In brief, 0.5 mL of fecal culture samples that were collected at each fermentation time point were centrifuged at 12,000 rpm for 5 min and then filtered through a 0.22 μm filter membrane. Each prepared sample (0.2 mL) was determined on a Polaris Amide-C_18_ column (Agilent, Palo Alto, CA, USA) at 30 °C. The mobile phase was composed of 20 mM KH_2_PO_4_ solution (pH adjusted to 2.5 by 30 mM H_3_PO_4_) (A) and methanol (B), and the detection wavelength was 210 nm.

### 2.10. Statistical Analysis

All the experimental data were performed in triplicate (*n* = 3) and exhibited as the means ± SD. Statistical analysis was conducted by SPSS version 20.0 (SPSS Inc., Chicago, IL, USA) and significant differences were analyzed by one-way analysis of variance (ANOVA) and Tukey’s multiple-range test. The *p* value < 0.05 indicated a significant difference.

## 3. Results

### 3.1. In Vitro Digestibility of LHEPS and LPEPS

In this study, the in vitro digestibility of LHEPS and LPEPS was investigated by stimulated saliva and GSI digestion. Figure 1 shows that the retention times of these two EPS fractions were unchanged, indicating that LHEPS and LPEPS were resistance to the simulated digestive system and could reach the large intestine with structural integrity. As for LHEPS, there were no dissociative oligosaccharides or monosaccharides in the digestion medium (Figure 1A), and the average Mw was the same (5.41 × 10^5^ Da) before and after digestion based on calibration with various standard T-series dextrans. In contrast, a minor peak a (Figure 1B) appeared after saliva and GSI digestion, implying that LPEPS released few oligosaccharides and/or monosaccharides (<5%) in the digestion medium. Besides, the above results also indicated that after digestion, LPEPS’s glycosidic bonds were partially broken and a few monosaccharides were generated, which led to a slight decrease in Mw (native LPEPS: 1.20 × 10^6^ Da) but with no appreciable impact on the repetitive structure units and polysaccharide’s main structure.

### 3.2. Effects of LHEPS and LPEPS on Probiotic and Enteric Bacteria Population

In this study, the impact of LHEPS and LPEPS on the specific probiotic and enteric bacteria in gut microbiota was investigated by employing the FISH technique to determine the population of the test groups. Figure 2 shows the results of anaerobic incubation with LHEPS at 12 h by FISH. The bacterial populations after in vitro fermentation at each time point that were supplemented with different purified ESP fractions are listed in Table 1. On the whole, the bacteria quantities of all the groups increased with a rapid growth in the early stage and the changes showed no obvious dependence on different EPS fractions. As for the growth of the total bacteria, the stimulating effect of LHEPS was superior than LPEPS at 48 h fermentation (9.78 and 9.48 log_10_ cells/mL, respectively). The bifidobacterial population (Bif164) experienced a significant increase (*p* < 0.05) in the anaerobic fermentation that was supplemented with LHEPS and LPEPS at 6 h, along with an insignificant increase during 6 h and 24 h, and a downward tendency at 48 h of fermentation (Table 1). Besides, the bifidobacterial populations that were stimulated by LHEPS and LPEPS at 48 h fermentation were comparable with those that were stimulated by α-glucan, β-glucan, and inulin. To be specific, the value of the bifidobacterial group population increased significantly (1.28–1.37 log_10_ cells/mL, *p* < 0.05) following fermentation of the LHEPS and LPEPS (from 7.11 log_10_ cells/mL to 8.39 and 8.48 log_10_ cells/mL, respectively). The changes of *Lactobacillus*/*Enterococcus* population (Lab158) were also investigated, and a substantial growth was observed for all the substrates during 48 h of fermentation. Comparing all the carbon sources that were tested to the control group, the growth trends of *Lactobacillus*/*Enterococcus* numbers were similar in the presence of LHEPS and β-glucan. As for LPEPS, the stimulation level of *Lactobacillus*/*Enterococcus* population was comparable with that of inulin. *Streptococcus*/*Lactococcus* numbers showed an insignificant increase with all the substrates at 24 h of incubation and a decline at 48 h. Precisely, the population of *Streptococcus/Lactococcus* reached the highest value at 12 h fermentation when stimulated by LHEPS and β-glucan, whereas LPEPS, α-glucan, and inulin increased the *Streptococcus*/*Lactococcus* population to the highest point at 24 h fermentation.

There were three groups of bacteria (*Clostridia*, *Bacteroides*, and *Escherichia coli*) that were chosen as undesirable objects in this study. As shown in Table 1, the numbers of *Clostridia* (His150) increased insignificantly in 6 h fermentation, but a downtrend was apparent after 6 h of fermentation in all cases, which were similar with previous results using the exopolysaccharide from a medicinal fungus as the sole carbon source [33]. *Bacteroides* dominated the bacterial populations at inoculation time (8.20 log_10_ cells/mL) and experienced a gradual increase (0 h to 24 h) with an insignificant decline (24 h to 48 h). Generally, all the EPS fractions showed no significantly different impact on *Bacteroides* population. Furthermore, the population of *Bifidobacterium* was almost equal to that of *Bacteroides* after 48 h fermentation, indicating a notable change of bacterial population distribution in human gut microflora. *Escherichia coli* (EC1531) populations showed a similar trend in all the tested groups with a gentle increase compared to the blank control.

The relative changes in the population size of different bacteria groups after 6 h, 12 h, 24 h, and 48 h of incubation that were supplemented with EPS fractions are summarized in Figure 3. On the whole, the population size of “beneficial” bacteria presented a rising tendency during the entire fermentation period, whereas that of the “less desirable” groups declined (except *Escherichia coli*). To be specific, inulin exhibited the most significant prebiotic effect followed by α-glucan after 6 h fermentation and followed by LHEPS after 12 h and 24 h fermentation. With further incubation, LHEPS showed the most significant promoting effect followed by LPEPS at 48 h fermentation. The population size of both *Bacteroides* and *Clostridia* decreased after 48 h fermentation. For all the bacterial groups that were monitored, the alternation of the population size was most noticeable with the addition of inulin.

### 3.3. Selectivity Index (SI)

To obtain the quantitative changes in the critical bacterial groups during fermentation and to compare the influence of the different EPS on the selectivity of fermentation, an SI value was determined according to previous literature [32]. As shown in Figure 4, the highest SI values were obtained in the present of inulin at 6 h, 12 h, and 24 h fermentation (18.79, 19.1, and 13.88, respectively). Notably, LHEPS achieved the greatest value of SI score (8.86) at 48 h, which was even higher than the relevant value of the widely recognized prebiotic inulin. Conversely, the samples with α-glucan as a carbohydrate source scored the lowest SI values at the three fermentation time points (6, 12, and 24 h), which were caused by low populations of bifidobacteria and lactobacilli and a rise in the number of *Bacteroides*. Additionally, LPEPS scored a higher SI value at 12 h and 24 h fermentation than that of LHEPS because of the risen populations of bifidobacteria and lactobacilli. Noticeably, LPEPS showed higher SI values than α-glucan and β-glucan due to significant growths and better activities of bifidobacteria and lactobacilli at 12 h and 24 h fermentation.

### 3.4. SCFAs Production during Fermentation

The contents of SCFAs and lactic acid during in vitro fermentations at each sampling time point are shown in Table 2. All the four SCFAs and lactic acid were produced significantly (*p* < 0.05) with the addition of each EPS fraction as compared to the control and attained the greatest values at 48 h fermentation. It was worth mentioning that α-glucan gave the maximum values of all kinds of SCFA as well as total acids of 125.06 mM even higher than that of inulin (120.66 mM). Compared to LHEPS, group of LPEPS contained a greater concentration of total SCFA and reached the highest level of 110.86 mM after 48 h fermentation. Acetic acid, lactic acid, and propionic acid were obviously the most predominant components that were generated during the fermentation. Besides, the results revealed that the levels of acetic, propionic, and lactic acid presented a sustainable increase throughout the fermentation, while the concentration of formic and butyric acid started to be detectable after 6 h of fermentation. The ultimate yields of butyric acid, which ranged from 4.29 to 5.38 mM during 48 h fermentation, presented no significant differences in all the substrate cultures. In this study, LPEPS achieved a higher level of all kinds of SCFAs than LHEPS at 48 h fermentation: 9.29 mM and 9.16 mM (formic acid), 35.74 mM and 35.20 mM (lactic acid), 45.91 mM and 44.65 mM (acetic acid), 15.04 mM and 14.84 mM (propionic acid), and 4.88 mM and 4.29 mM (butyric acid), respectively. Overall, LHEPS and LPEPS were highly related to α-glucan, β-glucan, and inulin according to the SI scores and mean SCFAs production.

## 4. Discussion

Carbohydrates that escape from human enzymes digestion and gastro-intestinal absorption and stimulate the growth or metabolic activity of beneficial microbes in the colon can be qualified as a prebiotic [34,35]. During the digestion period, α-amylase in saliva decomposes carbohydrates initially because of its hydrolyzation capacity towards starchy foods and some oligosaccharides [36]. After then, the acidic pH in stomach and bile salts in small intestine may influence the structures of polysaccharides. The results indicated that LHEPS from *L. helveticus* LZ-R-5 and LPEPS from *L. pentosus* LZ-R-17 were tolerant to saliva and GSI digestion and hardly degraded by these digestive enzymes. Therefore, it was expected that both of LHEPS and LPEPS could reach the gut with intact forms for colonic fermentation. Additionally, LHEPS showed a lower digestibility value than that of LPEPS, which could be attributed to the higher content of β-linkages in LHEPS [20,21]. Our previous study also reported that expolysaccharides from *L. delbrueckii* ssp. *bulgaricus* SRFM-1 reduced slightly in Mw after in vitro digestion, which was consistent with these results [27]. Besides, a galactan exopolysaccharide that was produced by *Weissella confusa* KR780676 presented a loss of 1.2% under intestinal juice hydrolysis and the same conditions occurred for the standard FOS [36]. Previous studies also concluded some elements that influence the digestibility of EPS, including source, molecular weight, and ratio of α- to β-linkages [37,38]. Therefore, the differential ability of LHEPS and LPEPS to modulate the intestinal microbiota and their potential prebiotic properties were further explored.

The normal balance between the gut microbiota and the host have been responsible for maintaining a healthy gastrointestinal tract and protecting against enteropathogens. Notably, over 400 different varieties of probiotic bacteria are responsible for inhibiting the growth of harmful bacteria and supporting a healthy digestive system [39]. According to the Guidelines for the Evaluation of Probiotics in Food promoted by the Food and Agriculture Organization of the United Nations and the World Health Organization (FAO/WHO) in 2002, necessary in vitro tests that mimic the hostile gut environment are recommended for screening potential probiotic strains, including resistance to gastric acidity, bile acid resistance, bile salt hydrolase activity, adherence to mucus and/or human epithelial cells, antimicrobial activity against potential pathogenic bacteria, and ability to reduce pathogen adhesion to surfaces [39]. As a complex dynamic ecosystem, the human gut microbiota is composed of a huge diversity of bacterial species and strains that can degrade various dietary carbohydrates [29]. Thus, in order to reveal the potential prebiotic effect of new oligosaccharide compounds, the evolution of the mixed bacterial population in the presence of LHEPS and LPEPS was investigated via in vitro fermentation of human intestinal microbiota. Previous studies reported that a prebiotic is capable of altering the colonic microbiota of the host toward a healthier composition. Specifically, selective stimulation of bifidobacteria (bifidogenesis) was considered a prebiotic effect, while nowadays a prebiotic also evokes a net of health benefits [5]. Therefore, the SI score was adopted to reflect the general quantitative comparative changes in “beneficial” bacteria populations (*Bifidobacteria*, *Lactobacillus*/*Enterococcus* spp., and *Streptococcus/Lactococcus* spp.) and ‘undesirable’ ones (*Bacteroides-Prevotella* group, *Clostridium histolyticum* group, and *Escherichia coli*). *Bifidobacterium* and *Lactobacillus* are common inhabitants of the human intestine and are important groups of gut commensals, which have beneficially affected human health through different mechanisms including strengthening the intestinal barrier, prevention from diarrhea and microbial pathogen infections, cholesterol and cancer risk reduction, and modulation of the immune response [40,41]. The value of the bifidobacterial group population increased significantly following fermentation of the LHEPS and LPEPS, which constituted a major shift toward a healthier composition in the gut microbiota since the enhancements exceeded 0.5–1.0 log_10_ cells/mL [42]. Therefore, the LHEPS and LPEPS would be considered as bifidus factors with greater bacterium levels than those that were activated by inulin (1.24 log_10_ cells/mL) [43]. As for the undesirable bacteria populations, although most of the Clostridia have a commensal relationship with the host, some Clostridia groups possess pathogenic species, such as *Clostridium perfrigens* and *Clostridium tetani*, which are members of Clostridium cluster I [44]. *E. coli* is also a versatile population including harmless commensal, probiotic strains as well as frequently deadly pathogens. In this study, the in vitro fermentation of all kinds of prebiotics did not cause a significant decrease of the *E. coli* group since it is a complex population with diverse species and vast quantity [45]. The results showed that LPEPS scored a higher SI value at 12 h and 24 h fermentation than that of LHEPS due to the increased populations of bifidobacteria and lactobacilli. Studies indicated bifidobacteria and lactobacilli are the predominant bacteria in infants and their population experiences a substantial decrease as humans age [38,46]. The alternation of the “beneficial” gut microbial population size is widely believed to be associated with human health. Besides, LHEPS achieved the greatest value of SI score at 48 h fermentation, even higher than that of inulin. Overall, changes in the bacteria population are mainly attributed to the selective utilization of supplied substrates by the gut microflora and cross-feeding effect by certain bacteria products [47]. Our previous studies also revealed that both of LHEPS and LPEPS presented immunostimulatory activity [20,21]. Therefore, these two EPS fractions could be regarded as prebiotic ingredients based on their primary chemical structures, immunomodulatory activity, the in vitro GSI digestion experiments, and prebiotic activities [20,21]. Firstly, glycosidic linkages in the structures facilitated the digestion tolerance of these EPS fractions and ensured their structural integrity until further utilization by colonic bacteria. Compared with LPEPS containing β-(1→4) linkages, LHEPS with β-(1→3) linkages presented a slightly higher SI value at 6 h fermentation, which was consistence with previous studies that reported that oligosaccharides with β-(1→3) linkages were selectively utilized for bifidobacteria and could be cleaved faster by β-galactosidase that was produced from colonic bacteria than β-(1→4) ones [29]. Additionally, LPEPS showed higher SI values than α-glucan and β-glucan at 12 h and 24 h fermentation, which may be attributed to its higher ratio of β- to α-linkages and synergistic effect by β-glycosidases and/or β-galactosidases working on both terminal linkages [27]. Overall, the chemical structures including monosaccharide composition, glycosidic linkages, and conformation of links between monosaccharides might influence the SI score and prebiotic activity of EPS fractions.

SCFAs that were derived from gut microbial fermentation of indigestible polysaccharides are crucial for intestinal health and are involved in the crosstalk between the gut and peripheral tissues. They affect various physiological processes, including activating G-coupled-receptors directly, improving glucose homeostasis and insulin sensitivity, and serving as energy substrates [48]. The notable acetic acid that was produced during the fermentation of all the substrates was consistent with a previous study that implied that a higher level of acetic and lactic acid was related with the increase of *Lactobacillus* strains since they inhibited the growth of bacterial pathogens by producing metabolites such as acetic and lactic acid to lower the pH [49]. Butyric acid usually provides energy for intestinal microorganism, reduces the oxidative stress, and increases the anti-inflammatory cytokine liberation [34]. All of the cultures produced considerable amounts of butyric acid. Currently, several reports have pointed out the close connection between relevant groups of bacteria with the production of the main SCFA resulting from saccharolytic fermentation [50]. As mentioned above, it has been confirmed that acetic acid is one of the major end-products of *Bifidobacterium* and *Lactobacillus* fermentation [51]. Additionally, butyric acid is probably related to the high abundance of *Megasphaera* by utilizing the galactose and galacturonic acid of polysaccharides [52]. Moreover, *Bacteroides-Prevotella* is known as the main producer of propionic acid [53]. As a whole, the high amount of SCFAs that were achieved by LHEPS and LPEPS fractions may provide various health benefits such as anti-cancer, anti-obesity, and anti-diabetic effects.

## 5. Conclusions

In summary, the regulatory effects of LHEPS from *L. helveticus* LZ-R-5 and LPEPS from *L. pentosus* LZ-R-17 on the gut microflora and SCFAs production have been evaluated by in vitro GSI digestion and fermentation in comparison with α-glucan, β-glucan, and inulin. Both of LHEPS and LPEPS showed no or few free oligosaccharides or monosaccharides that were released throughout the GSI digestion period, indicating the subsequent successful utilization by intestinal microbes. LPEPS exerted stronger promoting activity for the proliferation of *Bifidobacterium* and *Lactobacillus* than LHEPS with higher contents of SCFAs following human gut microbiota fermentation. Besides, LPEPS also showed higher SI values due to the risen populations of bifidobacteria, lactobacilli, and lactococci, which were even greater than α-glucan and β-glucan. Collectively, these results suggested that EPS that was derived from *L. helveticus* LZ-R-5 and *L. pentosus* LZ-R-17 would be a candidate as a potential prebiotic in the functional food industry. Further in vivo studies are necessary to assess the systemic metabolic health functionalities of these EPS fractions.

## Data Availability

The data that are presented in this study are available on request from the corresponding author.

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
