# Peer review of "Prebiotic Properties of Exopolysaccharides from Lactobacillus helveticus LZ-R-5 and L. pentosus LZ-R-17 Evaluated by In Vitro Simulated Digestion and Fermentation"

_foods, 2022, doi:10.3390/foods11162501_

Round 1
Reviewer 1 Report
Manuscript ID: foods-1847912
Type of manuscript: Article
Title: Prebiotic properties of exopolysaccharides from Lactobacillus helveticus LZ-R-5 and L. pentosus LZ-R-17 evaluated by in vitro simulated digestion and fermentation
Authors: Mengjia Xu, Zhi Li, Xiaogan Zhao, Wei Li *
Overall, the paper is well-structured and richly illustrated. Additionally, this study represents an interesting contribution to the field. However, the manuscript should be revised. To the authors I have the following questions:
- Abstract (Line 36): SCFAs should be described: “short-chain fatty acids”
- Lines 99-104: please, to give more information about the composition of culture medium, nutrients used, pH…
- Line 111: trademark, country… of the anion exchange column used.
- Lines 112-118: much more information about the methods high-performance liquid chromatography, Fourier-transform infrared spectroscopy, nuclear magnetic resonance spectroscopy, phenol-sulfuric acid method and the Bradford method should be shown.
- Lines 119-120: please, to explain in detail the methods used to determine the total uronic acid content and sulfate group content.
Author Response
1. Abstract (Line 36): SCFAs should be described: “short-chain fatty acids”.
Response: Thank you very much for your nice caution. We agree with your assessment and have revised the corresponding sentences in Introduction (Line 36).
2. Lines 99-104: please, to give more information about the composition of culture medium, nutrients used, pH…
Response: Thank you very much for your valuable comments. Additional information about the source of culture medium have been supplemented in Section 2.2.
3. Line 111: trademark, country… of the anion exchange column used.
Response: Thank you very much for your nice caution. More information about the anion exchange column including trademark, country and source have been supplemented in Section 2.2.
4. Lines 112-118: much more information about the methods high-performance liquid chromatography, Fourier-transform infrared spectroscopy, nuclear magnetic resonance spectroscopy, phenol-sulfuric acid method and the Bradford method should be shown.
Response: Thank you very much for your valuable comments. We agree with your suggestions and more information about the the methods of high-performance liquid chromatography, Fourier-transform infrared spectroscopy, nuclear magnetic resonance spectroscopy as well as phenol-sulfuric acid method and the Bradford method have been supplemented in Section 2.2.
5. Lines 119-120: please, to explain in detail the methods used to determine the total uronic acid content and sulfate group content.
Response: Thank you very much for your nice caution. We agree with your assessment and have explained in detail about the the methods used to determine the total uronic acid content and sulfate group content in Section 2.2.

Reviewer 2 Report
This is an interesting study and the authors have done extensive experiments. Furthermore, the results and discussion are well represented.
1. Line 106-109, please rephrase the sentence to make it more meaningful.
2. Line 120. Please explain a little bit about the "previous methods"
3. Line 131. The symbol of the degree to be corrected.
4. Line 135. The meaning of "filtrated" is not clear. "Filtrated" or "filtered"?
5. Line 282. Explain the reason why E. coli colony was not declined?
Author Response
1. Line 106-109, please rephrase the sentence to make it more meaningful.
Response: Thank you so much for your nice reminder. In our resubmitted manuscript, the sentences in Line 106-109 have been rephrased and marked in red font.
2. Line 120. Please explain a little bit about the "previous methods".
Response: Thank you very much for your valuable comments. We agree with your assessment and have explained in detail about the the methods used to determine the total uronic acid content and sulfate group content in Section 2.2.
3. Line 131. The symbol of the degree to be corrected.
Response: We were really sorry for our careless mistake. Thank you so much for your nice reminder. In our resubmitted manuscript, the symbol of the degree in Section 2.3 have been carefully checked and revised. In addition, the full manuscript have been carefully checked again and other writing mistakes have been also corrected and marked in red font.
4. Line 135. The meaning of "filtrated" is not clear. "Filtrated" or "filtered"?
Response: Thank you so much for your nice reminder. In our resubmitted manuscript, the word "filtrated" in Section 2.3 have been checked and revised into "filtered".
5. Line 282. Explain the reason why E. coli colony was not declined?
Response: We sincerely appreciate the valuable comments. Recent study have shown that Escherichia coli is one of the first colonizers of the gut after birth, which is also a versatile population including harmless commensal, probiotic strains as well as frequently deadly pathogens. In adults, E. coli remains the predominant aerobic organism in the intestine and the peaceful relationship benefits both partners from the interaction as commensalism. Moreover, commensal E. coli uses its host for a constant source nutrient supply, protection, transport, and dissemination (reference 1 as below). In our experiment, we chose EC1531 probe to specifically detect for most species of E. coli (reference 2 as below). Since the SI value represents a comparative relationship between the growth of ‘beneficial’ fecal bacteria and ‘undesirable’ ones, related to the changes of the total number of bacteria. In this study, E. coli group was regarded as typical undesirable strains. However, it was not necessary that the in vitro fermentation of prebiotics would cause significant decrease of E. coli group since it is a complex population with diverse species and vast quantity mentioned above. In our revised manuscript, we have explained the corresponding reason of the upward trend of E. coli colony in Discussion.
[1] Secher, T.; Brehin, C.; Oswald, E. Early settlers: which coli strains do you not want at birth? Am. J. Physiol. Gastrointest. Liver Physiol. 2016, 311(1), G123-G129.
[2] Franks, A.H.; Harmsen, H.J.; Raangs, G.C.; Jansen, G.J.; Schut, F.; Welling, G.W. Variations of bacterial populations in human feces measured by fluorescent in situ hybridization with group-specific 16S rRNA-targeted oligonucleotide probes. Environ. Microbiol. 1998, 64(9), 3336-3345.

Reviewer 3 Report
This manuscript uses the latest information and defined the mechanisms of absorption and utilization of prebiotics from LAB in the buccal and gastrointestinal tract of human body. However, it could be considered for publication after conducting a revision. I am hereby returning the paper to you, pointing out comments to the authors:
Materials and Methods
L103, “optimal inoculum concentration was 4%”, what lower or higher percentage could result?
L103, provide the source of milk (cow, goat, camel)
L 145-148, define PMP
L155-157, clarify if volunteers were consuming any dairy products contain probiotics (for example Yakult®, or Activia®)
Scheme 1. It is confusing, what arrows represent?
Results and Discussion
L254-268 and L317, the words increased, moderate, highest, slight have no statistical meaning, make sure to report the significant or insignificant levels of the results similar to those reported in L249, 273, 282.
Tables should be self-explanatory, define all abbreviations, in addition significant levels are written as upper superscript.
L372-373, To support this statement, Add the main currently used in vitro tests for the study of probiotic strains according to FAO/WHO, from table 1 article below
FAO/WHO (Food and Agriculture Organization of the United Nations and World Health Organization) (2002) Guidelines for the evaluation of probiotics in food.
Author Response
Materials and Methods
1. L103, “optimal inoculum concentration was 4%”, what lower or higher percentage could result?
Response: Thank you so much for your valuable comments. As reported in our previous studies (references below), factors affecting cell growth and EPS production were investigated using one-factor at a-time method. To find out the optimum inoculum concentration for EPS production, flask cultures were inoculated with 3%, 4% and 5%. The number of active bacteria was evaluated by incubating with MRS agar medium at 37 °C for 48 h and the EPS was collected by centrifugation at 12,000 rpm for 15 min. The EPS pellet was dissolved in distilled water, dialyzed, concentrated, lyophilized and weighed. The results showed that the inoculum concentration impact the EPS yield insignificantly. Precisely, inoculum concentration of 3% would extend the incubation time to reach the similar total bacteria number of 4% inoculum concentration, whereas the inoculum concentration of 5% would waste the bacterial seed. Thus, we chose 4% as the optimal inoculum concentration.
[1] Li, W.; Ji, J.; Rui, X.; Yu, J.; Tang, W.; Chen, X.; Jiang, M.; Dong M. Production of exopolysaccharides by Lactobacillus helveticus MB2-1 and its functional characteristics in vitro. LWT-Food Sci Technol. 2014, 59(2), 732-739.
[2] You, X.; Li, Z.; Ma, K.; Zhang, C.; Chen, X.; Wang, G.; Yang, L.; Dong, M.; Rui, X.; Zhang, Q.; Li, W. Structural characterization and immunomodulatory activity of an exopolysaccharide produced by Lactobacillus helveticus LZ-R-5. Polym. 2020, 235, 115977.
[3] You, X.; Yang, L.; Zhao, X.; Ma, K.; Chen, X.; Zhang, C.; Wang, G.; Dong, M.; Rui, X.; Zhang, Q.; Li, W. Isolation, purification, characterization and immunostimulatory activity of an exopolysaccharide produced by Lactobacillus pentosus LZ-R-17 isolated from Tibetan kefir. J. Biol. Macromol. 2020, 158, 408-419.
2. L103, provide the source of milk (cow, goat, camel)
Response: Thank you so much for your nice caution. In our resubmitted manuscript, we have provided the source of milk in Section 2.2. In detail, the pasteurized milk was sourced from cow in Bright Dairy & Food Co., Shanghai, China.
3. L 145-148, define PMP
Response: Thank you so much for your valuable comments. PMP is 3-methyl-1-phenyl-2-pyrazolin-5-one obtained from Sigma-Aldrich Chemical Co. (St. Louis, MO, USA). In our resubmitted manuscript, we have supplemented the definition of PMP in Section 2.1.
4. L155-157, clarify if volunteers were consuming any dairy products contain probiotics (for example Yakult®, or Activia®)
Response: Thank you so much for your nice reminder. In our resubmitted manuscript, we have clarified that the volunteers had not consumed any dairy products containing probiotics for at least half a year before participating in this trial in Section 2.5.
5. Scheme 1. It is confusing, what arrows represent?
Response: Thank you so much for your nice caution. In our resubmitted manuscript, we have revised the picture of Scheme 1 and replaced the arrow by “Sequence (from 5’ to 3’). The 16S rRNA-targeted oligonucleotide probes were single stranded DNA sequences and the Cy3 dye were labeled at their 5’ end.
Results and Discussion
L254-268 and L317, the words increased, moderate, highest, slight have no statistical meaning, make sure to report the significant or insignificant levels of the results similar to those reported in L249, 273, 282.
Response: Thank you very much for your valuable comments. We agree with your assessment and have revised the corresponding words into significant or insignificant in L254-268 and L317.
1. Tables should be self-explanatory, define all abbreviations, in addition significant levels are written as upper superscript.
Response: Thank you very much for your valuable comments. In our resubmitted manuscript, all abbreviations have been defined in Tables (Section 3.2 and 3.4) and marked in red font, the significant levels have been revised as upper superscript.
2. L372-373, To support this statement, Add the main currently used in vitro tests for the study of probiotic strains according to FAO/WHO, from table 1 article below
FAO/WHO (Food and Agriculture Organization of the United Nations and World Health Organization) (2002) Guidelines for the evaluation of probiotics in food.
Response: Thank you very much for your valuable comments. According to Guidelines for the Evaluation of Probiotics in Food promoted by Food and Agriculture Organization of the United Nations and World Health Organization (FAO/WHO) in 2002, necessary in vitro tests that mimic the hostile gut environment are recommended for screening potential probiotic strains, including resistance to gastric acidity, bile acid resistance, bile salt hydrolase activity, adherence to mucus and/or human epithelial cells, antimicrobial activity against potential pathogenic bacteria and ability to reduce pathogen adhesion to surfaces. To support the statement in L372-373, we have revised the corresponding sentences in Discussion and marked in red font (L422-430).
